# Enhancing Diversity and Accuracy in Personalized Tag Recommendations: A Hybrid Semantic and Contextual Analysis Approach

## Abstract

This paper introduces HYCOMB, a cascading Hybrid model that innovatively integrates Collaborative Filtering (CF), Content-Based Filtering (CB), and Context-Aware (CA) methods to address the challenge of data sparsity in tag recommendation systems. Unlike traditional models that rely heavily on user-item interactions, HYCOMB enhances recommendation diversity and interpretability by utilizing semantic clustering in CF to extract and analyze user sentiment from tags, adding a layer of nuanced understanding often missing in conventional systems. The CB component advances this by applying sophisticated NLP techniques to refine these recommendations based on item attributes, while the CA component incorporates movie synopses for deeper contextual understanding. Developed and tested using the MovieLens 20M dataset, our model demonstrates significant outperformance over baseline methods in terms of precision and recall, achieving scores of 0.813 and 0.364 respectively. Further, a newly introduced Overall Total Similarity metric that underscores its ability to deliver relevant and diverse recommendations. HYCOMB's strategic amalgamation of CF, CB, and CA not only mitigates the effects of sparse data but also improves the precision and diversity of tag recommendations, reflecting a more accurate alignment with user preferences.

## 1 Introduction

Tag recommendation systems are pivotal in organizing and personalizing content on various online platforms, especially within social media and content management systems. These systems leverage user-enabled tagging, where individuals freely assign tags or labels to describe content in a personalized manner, thereby enhancing user interaction. Tags serve as a critical form of User Generated Content, improving search accuracy, content management, and user engagement, which are essential for platform dynamics (Iferroudjene et al., 2023; Najafabadi et al., 2019b). Recommendation algorithms harness these tags to understand user preferences and generate personalized recommendations, thus guiding users toward content that resonates with their interests and enhancing content discoverability. The primary methodologies employed are collaborative filtering (CF), content-based (CB) filtering, and context-aware (CA) strategies, each presenting unique benefits and inherent limitations that could impact the efficacy of recommendations (Casillo et al., 2023; Najafabadi et al., 2017; Najafabadi & Mahrin, 2016) .

CF algorithms analyze user behavior data to identify patterns and suggest tags based on shared preferences. For example, users tagging "outdoor activities" might receive recommendations for "hiking" or "camping." Despite its utility, CF faces challenges with data sparsity, reducing the quality of recommendations when user interactions are sparse (Casillo et al., 2023; Lonjarret et al., 2021; Najafabadi et al., 2017; Najafabadi & Mahrin, 2016) . CB approaches analyze the features of items—textual, visual, etc.—to suggest relevant tags. For instance, a recipe might be tagged as "vegetarian" or "pasta." While CB methods are effective, they can result in narrow recommendations that may not fully reflect broader user interests or ensure diversity due to limited item context (Karabila et al., 2023; Najafabadi et al., 2019a; Sun et al., 2015) . CA methods enhance recommendations by incorporating contextual elements such as the user's location, time of day, or device type. For example, a context-aware system might recommend workout-themed playlists when a user is jogging in the morning (Casillo et al., 2023; Najafabadi et al., 2017).

Despite advancements in hybrid models, effectively integrating user sentiment and contextual relevance remains a substantial challenge. This research draws on seminal works to address data sparsity and enhance the precision and relevance of recommendations through the systematic incorporation of sentiment analysis (Najafabadi, 2024; Karabila et al., 2023; Duan et al., 2022; Onan, 2023) .

The primary objective of this research is to develop and validate HYCOMB, a hybrid tag recommendation system that addresses data sparsity and improves the interpretability and diversity of recommendations. The main contributions of this paper are as follows:

- The model utilizes GloVe vectors for semantic analysis within the CF component. This approach not only aids in clustering but also enhances the sentiment analysis of tags, leading to more nuanced understanding and utilization of user-generated content. The algorithm employs cluster tagging to establish a semantic context, overcoming the challenge posed by non-sentiment tags such as "genre" or "director." By grouping tags based on their meanings, the model differentiates between sentiment-related tags and descriptive tags, crucially enhancing recommendation accuracy and relevance.

- The context-aware model component extracts and summarizes movie synopses from multiple sources (IMDb and TMDB), which is then used to enrich the movie context vectors. This inclusion of textual context allows for more precise recommendations based on the narrative content of the movies. The Context-Aware Model enhances accuracy and relevance by using movie synopses and genre information to create rich context vectors. This integration effectively addresses tag ambiguity and redundancy, refining recommendations to better align with actual movie content and user contexts.

- In the CB filtering pathway, our model applies Named Entity Recognition (NER), semantic clustering, unsupervised word sense disambiguation (WSD), and sentiment analysis to refine the content vectors. This multifaceted approach ensures that the tags are not only relevant but also contextually and emotionally aligned with the user preferences.

- The final recommendation combines insights from all three filtering methods, ensuring that the output is both diverse and personalized. This process utilizes a sophisticated algorithm to weigh and integrate the individual contributions from each filtering approach, resulting in high-quality recommendations. We evaluate the effectiveness of our proposed HYCOMB method through testing on the MovieLens, employing metrics such as precision, recall, F1-score, and Overall Total Similarity (OTS). Our experimental results demonstrate that HYCOMB consistently outperforms baseline methods, striking an optimal balance between precision and recall, and significantly enhancing the user experience with more relevant and contextually appropriate recommendations.

The organization of this paper is as follows: Section 2 discuss the methodologies employed in previous approaches, identifying their limitations, and outlining the gaps these limitations present within the field. In Section 3, we delve into a comprehensive exploration of each element of our approach. Section 4 showcases the assessment of our model, detailing the outcomes achieved through rigorous evaluation. Finally, We justify the achievement of results and limitations and summarize the key findings and contributions in Section 5.

## 2 RELATED WORK

### 2.1 OVERVIEW OF TAG RECOMMENDATION APPROACHES

CF is a widely used method in tag recommendation systems, often chosen to tackle the data sparsity issue. This challenge arises from the inherent structure of the user-item matrix, which records interactions between users and items based on previous ratings or tags. Typically, this matrix is sparse, with few non-null entries, as the number of items significantly exceeds the number of users.

User-based CF predicts ratings by identifying users with similar rating behaviors. These similar users form a basis for predictions, making CF particularly effective in the presence of sparse data. Furthermore, clustering has been successfully applied to enhance CF systems by grouping items based on semantic similarity. This approach creates a denser representation of preferences, improving the accuracy of similarity calculations within a sparse matrix. Hybrid models have demonstrated

promising results in addressing the data sparsity problem in recommendation systems. These models often merge matrix factorization techniques with neighbor-based methods, leveraging the strengths of one approach to offset the weaknesses of the other. For example, hybrid collaborative recommendation approaches combine collaborative filtering with matrix factorization and implicit user feedback (Xu et al., 2023; Xin et al., 2022; Zhang et al., 2020) .

Review-Based Matrix Factorization (RMF) is highlighted as a method that integrates review-based hybrid collaborative filtering with matrix factorization, specifically addressing data sparsity issues. RMF utilizes sentiment analysis to extract detailed product features and consumer opinions from reviews, constructing an item-topic rating matrix for unknown user rating predictions. This approach not only mitigates overfitting problems in sparse rating datasets but also harnesses textual reviews to enhance recommendation accuracy, demonstrating significant applicative value in product promotion (Duan et al., 2022).

Another effective hybrid method involves co-embedding item attributes and user ratings (Onan, 2023; Najafabadi et al., 2021; Yang et al., 2018), this approach integrates a rich set of item features to understand the semantic similarity between items, which is crucial when user-item interactions are sparse. By blending item attributes with user ratings, the model gains insights into user preferences and identifies which attributes are most influential in recommendations.

Previous techniques of fusion models used for recommending tags has attracted attention, the attempts have been proposed to improve the performance. The method hybrid CF with tags and time (Zhang et al., 2018) novelly combines user-based and item-based CF with utilizing tag information to calculate item or user similarity to address data sparsity. However, its genre tag may not capture nuances of movie content granularity and the cosine similarity for tag calculation may not show the weighted importance of tags. Another attempt hybrid CF with convolutional neural network model (Khan et al., 2021) combines traditional CF and deep learning methods to improve tag recommendation, it analyzes how CNN and CF model complement each other but doesn't analyze the correlation between tags for recommendation, although outperforms state-of-the-art methods in terms of recall, it does not perform well on other metrics especially when training data is sparse. Deep-learning methods are also used to build recommendation models, label-wise attention and adversarial training for tag recommendation (Wang et al., 2021) adopts BERT-based label-wise attention using adversarial training to fine-tune model performance but increases the chance of overfitting and poor generalization to unseen data. Graph-based model for knowledge enhanced tag-aware recommendation (Wang et al., 2022) is proposed with the use of auxiliary user tag knowledge to enhance user preference modeling, it can naturally address the sparsity issue with an additional attention-based graph that adaptively learns from heterogenous neighbours. The complexity of collaborative recommendation graph reduces transparency in prediction and require external sources for data to construct the auxiliary graph which may not be carried out in dataset in small scale.

The work presented in our paper is designed to address the challenges of data sparsity and enhance the personalization of tag recommendations. Unlike existing methods, which typically focus on singular aspects of recommendation systems, our approach synergistically leverages multiple dimensions of user and item data. We enrich traditional CF and CB models by incorporating sentiment analysis, adding an emotional context that captures more nuanced user preferences. Our context-aware component further tailors recommendations to users' environmental and temporal contexts, ensuring a dynamic and responsive recommendation system. This comprehensive model not only addresses the limitations of traditional hybrid models in handling sparse data but also significantly enhances the diversity and accuracy of recommendations, marking a pioneering advancement in the field of recommendation systems.

## 2.2 BASELINES

To ensure a robust evaluation of our proposed HYCOMB model, it is crucial to compare its performance against other established methods. This comparative analysis not only highlights HYCOMB's strengths and potential areas for improvement but also situates our contributions within the broader context of tag recommendation research. To evaluate our model's ability to leverage NLP techniques, compared with established embedding-based methods, deep learning context based method and graph-based model.These baselines were chosen for their relevance to our research aims and their recent contributions to the field of tag recommendation:

- **NLP with Machine Learning Embeddings (NLP_ML) (Najafabadi et al., 2021)**: This approach utilizes word embeddings to analyze relationships between words and target objects. Features for candidate terms are extracted using embeddings and refined through a skip-gram model, followed by a learning-to-rank method that prioritizes relevant tags.

- **Syntactic and Neighbourhood Tag Attributes (SYN_TAG, SYN_TAG) (Belém et al., 2019)**: This method integrates syntactic tag quality features into a Learning-to-Rank (L2R) system. It further enhances recommendations by expanding the neighbourhood of target objects based on syntactic patterns to improve tag relevancy.

- **Deep Learning and Multi-Objective Recommendation (TRDM) (Zuo et al., 2023)** : Employing a deep learning framework, this method operates in two stages: initial tag-based recommendations using deep neural networks, followed by multi-objective optimization to refine the recommendations based on accuracy and diversity.

- **Graph-based Model for Knowledge-enhanced Tag-aware Recommendation (TKGAT) (Wang et al., 2022)**: This model constructs a Collaborative Recommendation Graph (CRG) to capture the complex interplay between user-item interactions, tags, and auxiliary data. It utilizes a BERT-based label-wise attention mechanism to focus on different network elements, recommending tags based on a nuanced understanding of context and interactions.

## 3 Proposed Method

Figure 1 offers a visual representation of our hybrid recommendation model. By integrating CF, CB, and CA techniques, our approach aims to deliver more accurate and personalized tag recommendations. The process begins with data pre-processing, setting the stage for the generation of initial recommendations within the CF model. These preliminary recommendations are subsequently cascaded into the CB model for additional refinement. A crucial component of our model is the CA model, which produces a contextualized vector based on the item. This vector is essential for the word sense disambiguation task undertaken by the CB model. Word sense disambiguation is a technique in NLP to identify the intended meaning of a word in a given context particularly when the word has interpretations. For instance, the tag "apple" could refer to the fruit or the technology company, depending on the context provided by other tags and content. By accurately identifying the intended meaning, our model ensures that the recommendations are contextually relevant. Ultimately, the final tag recommendations are generated within the CB model, ensuring they are tailored to user preferences and contextually appropriate.

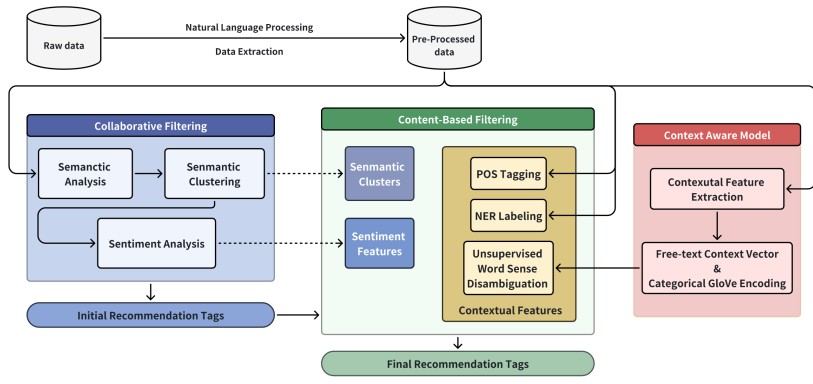

Figure 1: Architecture of HYCOMB recommendation model

### 3.1 Collaborative Filtering Based on Sentiment Analysis

In order to better generate candidate tags with utilizing advanced natural language processing, we propose collaborative filtering based on sentiment analysis. Initially, we process pre-processed tags along with a subset of user data. This involves conducting semantic analysis to generate clusters of similar tags, utilizing their GloVe vector representations. Subsequently, sentiment analysis is

performed on these clusters to evaluate the emotional context of the tags. Following this, we apply traditional CF to these sentiment-enriched tags to generate an initial set of recommendations. This set serves as the input for further refinement in the CB model. Here's how the process unfolds:

**Step 1:** Semantic analysis is the initial step where we convert tags into numeric representations using pre-trained GloVe vectors. This process enables the model to understand and interpret the meanings of tags based on their usage in specific contexts. Utilizing GloVe embeddings pre-trained on extensive datasets like Wikipedia allows us to capture complex semantic relationships without the overhead of training from scratch, ensuring our model quickly adapts to the nuances of movie-related content. **Step 2:** Following semantic analysis, tags are clustered to enhance the semantic precision in sentiment analysis. We employ a Hybrid Hierarchical-K-means++ Clustering approach, which optimizes centroid selection to improve cluster quality. This method effectively groups semantically similar tags, facilitating the aggregation of sentiments that are expected to be similar. The clustering process ensures that the tags within each cluster reflect a uniform sentiment, which simplifies the subsequent sentiment analysis steps. **Step 3:** After clustering, sentiment analysis is conducted using a fine-tuned pre-trained classification model. The model is initially fine-tuned on a subset of manually labeled tags, identified through active learning to maximize efficiency. Once fine-tuned, the model's accuracy is validated and then used to assign sentiment values to the remaining tags, ensuring comprehensive and precise sentiment mapping across the dataset. **Step 4:** In the final step, we construct a user-movie matrix where each entry reflects a sentiment value derived from user-tagged interactions. This matrix forms the basis of our collaborative filtering process, where sentiment similarity between users guides the recommendation system. By employing a K-Nearest Neighbours algorithm, we identify users with similar sentiment tendencies, allowing us to recommend movies that align with their expressed preferences.

Through this process, recommendations are tailored not only by suggesting movies but also by predicting tags that these users are likely to use based on their sentiment profile. The list of recommendations for each user is passed to the content-based filtering model. This provides a set of initial recommendations that is further enhanced and refined.

## 3.2 CONTENT-BASED REFINEMENT IN COLLABORATIVE FILTERING

The content-based (CB) filtering model enhances the recommendations from the CF approach by addressing data sparsity issues and leveraging tag-based features to improve recommendation quality. In this process, we utilize the same subset of users and tags processed in the CF model to ensure consistency and stability in the foundation of our recommendations. This alignment helps maintain coherence between the models. Five components contained in Content-Based Analysis: Named Entity Recognition (NER), Part of Speech (POS) Tagging, Semantic Clusters, Sentiment Features, and Unsupervised Word Sense Disambiguation (WSD).

• Based on WNUT 2017 taxonomy, we fine-tune the NER pre-trained model using MoDAL to fit our data. By assigning NER labels to tags, we can identify and classify named entities within tags to better understand their context and relevance.

• Employing POS tagging from spaCy model, we analyze the grammatical roles of tags, aiding in the precise interpretation of their semantics.

• Semantic Clusters and Sentiment Features are extracted from the previous CF model, which respectively provide group similar meanings and sentiments and deepen understanding of user preferences expressed through tags, thus enhancing the specificity of the recommendations.

• Critically, the model resolves ambiguities in tag meanings by utilizing a context-aware model. Unlike typical WSD applied at sentence or document levels, our model tackles single-word tags which lack surrounding lexical information. To effectively disambiguate these tags, we utilize context vectors created in the Context-Aware Model for each movie. For instance, the tag 'bat' could ambiguously refer to both an animal and sports equipment. By applying the context vector, 'bat' associated with a documentary about animals is accurately identified as the mammal, while in a baseball film, it is recognized as sporting equipment. This nuanced approach ensures that each tag is interpreted in its correct context, significantly enhancing the accuracy of our tag-based recommendations.

After the tags are fully processed and analyzed, the model refines the initial CF recommendations by comparing the semantic and sentiment-related features of user tags with those of the recommended

tags. This comparison employs kernel density estimation between context vectors and sentiment scores to assess similarity. The model outputs a list of tag recommendations that best match the user's preferences based on semantic similarity and sentiment alignment. These recommendations are tailored to enhance user satisfaction by closely aligning with individual user profiles and preferences.

This content-based filtering approach not only fills the gaps left by the CF model due to data sparsity but also adds a layer of depth to the recommendations by considering the nuanced meanings and sentiments of tags.

### 3.3 INTEGRATION OF CONTEXT IN THE HYBRID MODEL

In our hybrid recommendation model, context is critical for interpreting user-item interactions effectively, particularly within the social/entertainment domains characteristic of the MovieLens dataset. This model views the item, specifically movies, as essential contextual features. Tags, when used in isolation, can introduce ambiguity due to their varied meanings across different contexts. For instance, the same tag may convey different meanings when applied to different movies, especially since there are no restrictions on tag repetition. Three steps ensure movie-related data is integrated to establish a cohesive contextual framework for effective tag disambiguation and refined recommendation delivery.

**Step 1.** Identification of Contextual Features: Movies are designated as key contextual features due to their critical role in clarifying the meanings and relevance of tags. This strategy acknowledges that tags, devoid of contextual support, are prone to misinterpretations since their significances can shift dramatically across different films. **Step 2.** Creation of Context Vectors for Movies: We extract synopses from trusted sources to capture the thematic and narrative elements of each movie. Additionally, we construct weighted context vectors utilizing the retrieved synopses and movie genres. We employ GloVe for semantic richness and TF-IDF to emphasize contextually significant terms within the synopses, ensuring the vectors are both rich in semantic detail and contextually relevant. **Step 3.** Integration with Content-Based Model: Application in Word Sense Disambiguation (WSD): Within the content-based filtering model, these context vectors are pivotal for the WSD task. They furnish vital insights into the broader narrative and thematic contexts of the films, facilitating more accurate disambiguation of tag meanings relative to the movie genre and synopsis.

This approach not only boosts the specificity of our recommendations by ensuring tags are aligned with their corresponding movie contexts but also elevates the overall accuracy of the content-based model's outputs.

## 4 EXPERIMENTAL

### 4.1 EVALUATION DATASET

Our recommendation algorithm leverages the MovieLens 20M dataset from the GroupLens, which contains 20 million ratings and 465,000 user-generated tags from 138,000 users across 27,000 movies. We selected this dataset for its balanced trade-off between computational efficiency and tag diversity, offering optimal relevance and manageable computational demands. For our study, we primarily utilized the movies, tags, and links files from the dataset.

To comprehensively understand user tagging behavior and the inherent sparsity in our dataset, as illustrated in Figure 2, we analyzed the MovieLens dataset using several metrics and visualizations: **(a) Sparsity Calculation**: Sparsity in the CF model is determined by measuring the ratio of missing values to all interactions in the user-item matrix. Given that not all users interact with every available item, the matrix exhibits a high sparsity of 99.79%, confirming the anticipated sparse nature of user interactions. **(b) User-Movie Interactions**: Interactions between users and movies are visualized in top of Figure 6, with blue dots indicating tagging activity. Despite the appearance of uniform interaction across the dataset, the high sparsity value indicates that many interactions involve repeat tags by users on the same movie, rather than a broad engagement across the full item set. **(c) Distribution Analysis**: Distribution Analysis: Various metrics reveal a right-skewed distribution pattern. The interactions per movie show that most movies receive fewer than 20 tags, suggesting a concentration of tagging activity around a small subset of movies while the majority receive few

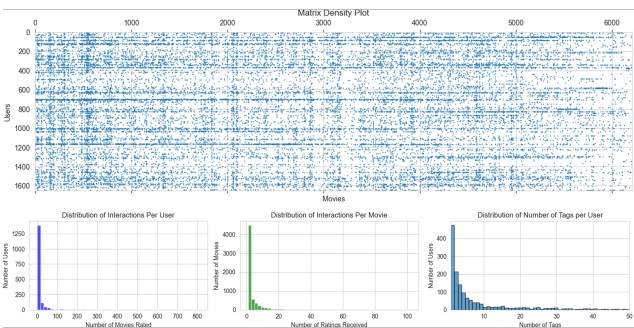 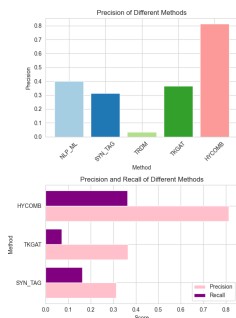

Figure 2: Description of Dataset                    Figure 3: Comparison Results

interactions. Similarly, the histogram of user interactions demonstrates that most users engage with only a few movies, supporting the observed sparsity in the user-item matrix. Additionally, the distribution of tag counts per use show that while a small number of users assign many tags, the majority contribute only a few, highlighting the sparse tagging behaviour among users.

These analyses underscore the challenges posed by data sparsity in CF models and help in tailoring our recommendation strategies to better predict and serve user preferences despite limited interactions.

## 4.2 EVALUATION METHODOLOGY

Our evaluation methodology is fully automated leveraging a subset of previously assigned tags as ground truth for assessing the recommendation accuracy, as described by Belém et al. (2019). With this approach, the subjectivity and cost of manually evaluating the top 10 suggested tags are avoided. We employ 5-fold cross-validation reserving 80% of each user's tags for training and using 20% as ground truth, with key evaluation metrics precision, recall, F1-score (F1) and Overall Tag Similarity (OTS). Let $A_u$ be the set of actual tags for user $u$. Let $R_u$ be the set of top 10 suggested tags for user $u$ generated by the method. Then we have $Precision = \frac{1}{n} \sum_{u=1}^{n} \frac{|R_u \cap A_u|}{|R_u|}$ reflecting the relevance, $Recall = \frac{1}{n} \sum_{u=1}^{n} \frac{|R_u \cap A_u|}{|A_u|}$ reflecting the coverage, and also $F1 = 2 \times \frac{(Precision \times Recall)}{(Precision + Recall)}$ accounting for both the relevance and completeness of the recommendations.

Specifically, overall tag similarity evaluates the diversity and relevance of recommendations across all users. Given by $OTS = \frac{1}{n} \sum_{u=1}^{n} AverageSimilarity_u$, it calculates the average similarity of the top 10 tags per user, assessing how well the recommended tags align with each user's tagging behavior. This score ranges from 0 to 1, with values closer to 1 indicating highly personalized and relevant recommendations.

## 4.3 EVALUATION RESULTS AND BASELINES COMPARISON

Our algorithms effectiveness was evaluated through a 5 cross validation method emphasizing precision, recall, F1 score and Overall Tag Similarity (OTS). The outcomes can be found in Table 1 displaying the metrics values for each fold. This table showcases the highest values in every fold with green denoting the highest scores and red representing the lowest ones. The average values for these metrics are presented in the last row.

• **Precision and Recall**: Illustrated in Table 1 the average precision and is 0.813 and 0.364 respectively. This indicates that the model is able to effectively exclude irrelevant tags and identify relevant ones, resulting in a fairly high precision score while with some reduction in recall. The precision and recall of each fold vary slightly, particularly the precision ranges from 0.800 to 0.832, and the recall ranges from 0.355 to 0.382, indicating that our model performs stably across five training-testing splits and shows a consistency across different partitions of data. • **F1-Score**: The average F1-score stands at 0.503, demonstrating a relative balanced trade-off between precision and recall. Given the data sparsity, an average score above 0.5 underscores the model's robust recommendation capabilities. • **Overall Tag Similarity (OTS)**: The OTS depicted by a pink bar, achieves a remarkable

Table 1: 5-Fold CV Results

| Fold | Precision | Recall | F1 Score | OTS |
|------|-----------|--------|----------|-------|
| 1 | 0.808 | 0.360 | 0.498 | 0.933 |
| 2 | 0.811 | 0.364 | 0.502 | 0.932 |
| 3 | 0.813 | 0.360 | 0.499 | 0.933 |
| 4 | 0.800 | 0.355 | 0.492 | 0.933 |
| 5 | 0.832 | 0.382 | 0.523 | 0.935 |
| **Average** | 0.813 | 0.364 | 0.503 | 0.933 |

Table 2: Baselines Comparison Results

| Baseline | Precision | Recall |
|----------|-----------|--------|
| **NLP_ML** | **0.400** | - |
| **SYN_TAG** | 0.314 | **0.162** |
| **TRDM** | 0.035 | - |
| **TKGAT** | 0.365 | 0.072 |
| **HYCOMB** | **0.813** | **0.364** |
| **Improve.** | 203% | 225% |

average score of 0.93. This metric underscores the algorithm's ability to recommend both diverse and relevant tags, enhancing user engagement by introducing new, yet contextually appropriate tags.

By integrating enhancements in sentiment, sematic and contextual analysis, our model excels in recommending higher-quality results and improve the coverage of topics that users are interested in. The high OTS score is particularly noteworthy, reflecting the model's effectiveness in enhancing user experience through personalized and relevant tag recommendations.

As detailed in Table 2, Figure 3 utilizes bar charts to visually contrast the performance metrics between our model (HYCOMB) and the established baselines. In the illustration of the precision metrics, our algorithm significantly outperforms all baselines, recording a precision of 0.813, which is more than double the highest precision achieved by any baseline method (NLP_ML at 0.400). This demonstrates our algorithm's superior ability to accurately identify relevant tags.

In comparison with the performance of three tag recommendation methods—SYN_TAG, TKGAT, and HYCOMB regarding precision and recall. With respect to recall metrics, HYCOMB not only continues to outclass all baseline methods but does so by a considerable margin. For example, the best-performing baseline, SYN_TAG, registered a recall of 0.162, whereas HYCOMB achieved a recall of 0.364, demonstrating an improvement of over twofold. This remarkable performance indicates our model's exceptional ability to retrieve a substantial proportion of relevant tags.

The graphical representations in these figures clearly underscore our algorithm's enhanced performance across all key metrics compared to the baselines. This superior performance underscores the effectiveness of our algorithm in providing comprehensive and accurate tag recommendations, significantly improving user experience by effectively aligning recommendations with user interests.

The algorithm's success stems from its sophisticated hybrid architecture that integrates high-dimensional GloVe embeddings with sentiment analysis, which allows for deeper semantic relationships and enhanced granularity of tag representations. Unlike the NLP with Machine Learning Embeddings (NLP_ML) method, which relies primarily on basic word embeddings and structural similarity, our model enriches tag recommendations by incorporating user sentiment, enabling personalized and context-aware recommendations through a Hybrid Hierarchical-K-means++ clustering approach. This method effectively captures the emotional relevance of tags, contrasting sharply with SYN_TAG's reliance solely on syntactic structures. By integrating user preferences and a broader range of tag features, our model excels in sparse data scenarios, surpassing SYN_TAG by employing a collaborative filtering framework augmented with sentiment analysis, thus generating more accurate tag recommendations. Additionally, our approach dramatically outperforms the Deep Learning and Multi-Objective Recommendation (TRDM) method, especially in environments characterized by high data sparsity. By eschewing deep learning networks in favor of a hybrid approach that combines clustering, content-based features, and proven recommendation methods, our model avoids the information loss typical of neural networks and effectively addresses data sparsity, enhancing both the precision and personalization of recommendations. Furthermore, our model surpasses the Graph-based Model for Knowledge-enhanced Tag-aware Recommendation (TKGAT) by leveraging advanced GloVe embeddings and context-aware tag assignment through Word Sense Disambiguation, enhancing semantic depth and reducing redundancy. The integration of item-based attributes and a context-aware movie synopsis enables our algorithm to distinguish between similarly tagged but contextually different movies, thus significantly boosting the relevance and diversity of the recommendations compared to TKGAT's static graph structure. This comprehensive integration and the nuanced handling of tag features substantiate our model's capability to deliver high-quality,

personalized tag recommendations, setting a new benchmark in the recommendation system landscape.

## 5 DISCUSSION

### 5.1 IMPLICATIONS

• **Innovative Integration of Hybrid Methods**: This research advances the theoretical underpinnings of recommendation systems by demonstrating how a cascading hybrid model, integrating CF, CB, and CA, can significantly mitigate issues of data sparsity. Our approach not only addresses the sparsity but also enhances the diversity and accuracy of recommendations through a nuanced synthesis of methodologies.

• **Innovations in Semantic Analysis and Sentiment Integration**: By utilizing advanced NLP techniques such as Named Entity Recognition, Part of Speech Tags, and sentiment analysis in conjunction with semantic clustering, our model contributes novel insights into the synthesis of textual features and user sentiment. This enhances the granularity and accuracy of tag recommendations, extending the theoretical understanding of tag-based systems.

• **Improved Recommendation Quality**: The model demonstrates high precision and recall in environments characterized by data sparsity, showcasing its practical efficacy in generating quality recommendations. This is crucial for platforms with limited user interactions, where conventional models typically falter.

• **Differentiation from Existing Work**: Our approach distinctively integrates sentiment analysis not just as a feature but as a pivotal element in the recommendation process, balancing it with other textual and contextual features to optimize tag recommendations. Unlike existing models that may either overlook the depth of sentiment analysis or fail to integrate it effectively with other features, our model ensures that sentiment enriches the recommendation process without overwhelming it. This balanced integration is particularly adept at handling the sparsity and diversity challenges in tag recommendation systems, setting our work apart from conventional methods.

### 5.2 LIMITATIONS AND FUTURE WORK

While our research demonstrates significant advancements in recommendation systems through a hybrid model, there are limitations that point to valuable directions for future work:

• **Topics Expansion**: Although our method has significantly improved recall score, there remains a gap with precision, indicating that the model is relatively conservative in recommendation. To further enhance our model's performance, we will investigate more effective combinations of NLP features to cover a broader range of relevant topics.

• **Domain Adaptation**: Our model has been exclusively tested with the MovieLens dataset. To assess its generalizability, future studies should apply the model across various domains, potentially revealing unique tagging behaviors and domain-specific nuances not present in the MovieLens context.

These identified areas not only underscore the need for continued refinement of our recommendation system but also open avenues for enriching its capabilities to meet the dynamic demands of modern users and diverse application scenarios.

### 5.3 CONCLUSION

In this study, we successfully demonstrated the efficacy of a hybrid recommendation system that intricately combines Collaborative Filtering (CF), Content-Based Filtering (CB), and Context-Aware models (CA). This innovative integration effectively surmounts the individual limitations inherent in each method by synergistically leveraging their strengths. Our model addresses the prevalent issue of data sparsity in CF through the implementation of semantic clustering using GloVe vectors, enhancing both the diversity and the quality of recommendations. Simultaneously, the CB model has been significantly advanced with the generation of rich textual features through techniques like Named Entity Recognition (NER), Part of Speech (POS) tags, and sentiment analysis, enabling a

profound understanding of content beyond the conventional scope of CB systems. Furthermore, the incorporation of context-aware strategies allows for precise tag recommendations by differentiating between semantically similar tags across varying contexts. The operational enhancements such as sentiment-infused clustering and balanced feature weighting in the CB model not only reduce computational demands but also ensure that recommendations are finely tuned to user preferences without overemphasis on any single feature. The culmination of these methodological advancements is a robust system that consistently delivers high precision and recall across sparse datasets, setting a new standard in the field of recommendation systems. Our research confirms that a well-designed hybrid model can significantly improve recommendation accuracy and user satisfaction, paving the way for future innovations in personalized content delivery systems.

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
