# OpenReview forum: "ENHANCING DIVERSITY AND ACCURACY IN PERSONALIZED TAG RECOMMENDATIONS: A HYBRID SEMANTIC AND CONTEXTUAL ANALYSIS APPROACH"
_ICLR.cc/2025/Conference — ICLR 2025 Conference Withdrawn Submission_

### Official Review · Reviewer_tnu7 · 2024-10-27

**Soundness:** 2
**Presentation:** 1
**Contribution:** 1
**Rating:** 3
**Confidence:** 3

**Summary:**

The paper introduces HYCOMB, a novel tag recommendation model that blends Collaborative Filtering (CF), Content-based Filtering (CB) and Context-aware (CA) techniques to resolve data sparsity. HYCOMB advances existing approaches by combining semantic information brought by integrated methods, such as sentiment analysis of tags, inferring item attributes by NLP techniques from CB or contextual information from CA. On MovieLens 20M dataset, HYCOMB remarkably outperforms comparable baselines. HYCOMB is not only capable of resolving data sparsity but also demonstrates more diverse recommendation as well as makes more accurate tag recommendations.

**Strengths:**

1. The proposed idea of integrating content-based and context-aware methods to improve collaborative filtering is sound. Semantic information, such as item attributes and contextual information, has a potential to resolve data sparsity problem of recommendation data.

2. The proposed model HYCOMB performs remarkably well on MovieLens 20M dataset, significantly outperforming comparable prior works.

**Weaknesses:**

1. The methodology is not sufficiently described. Various important aspects of the proposed method are missing, such as the recommendation algorithm in CF component, the mathematical description of integrating CB and CA into CF, input/output of the proposed model HYCOMB.

2. Experimental results are not convincing. First, only MovieLens 20M data is utilized, which is insufficient to evaluate the strength of the proposed method. Second, while the proposed HYCOMB includes multiple components such as CB, CA, their individual contributions are not studied. Third, efficiency of HYCOMB is not analyzed, making it difficult to understand the pros and cons of the proposed method.

3. Despite being mentioned frequently through out the main text that the goal is to resolve data sparsity, there lacks of substantial evaluation on whether the proposed method is capable of dealing with sparsity.

**Questions:**

1. Which recommendation algorithm is employed for CF component?

2. How CB and CA are integrated into CF? Please describe this mathematically.

---

### Official Review · Reviewer_qnwV · 2024-11-02

**Soundness:** 1
**Presentation:** 2
**Contribution:** 2
**Rating:** 3
**Confidence:** 4

**Summary:**

The paper propose HYCOMB, which combine Collaborative Filtering(CF), Content-Based Filtering(CB), and Context-Aware(CA) to sovle the challenge of data sparsity in tag recommendation system.

**Strengths:**

1.	The paper is well-written。

**Weaknesses:**

1.	This paper significantly lacks literature review, ignoring the latest advances in both recommender systems [4, 5] and natural language processing [6].
First, there is almost no references since fourth page of this paper, and the whole paper only contains around 20 references, which is extremely unprofessional for an academic paper, especially in the rapid-evolving domain of artificial intelligence and recommender systems. This lack of literature review makes me doubt the understanding comprehensiveness of this paper in this research field.
Second, this paper ignores the latest research advances. For example, the semantic embedding used in this paper is GloVe [1], which is quite old and weak. This paper ignores advances in the embedding models, such as LLM2Vec [2] and OpenAI’s text-embeddings-3 model [3]. Meanwhile, this paper even does not cite GloVe while using it.
2.	The motivation of this paper is unclear, resulting in the limited novelty. The reason for combining CF, CB, and CA is unknown. The correlation between the integration of these components and addressing the data sparsity is also questionable.
3.	The experiments in this paper are inadequate to prove the effectiveness of the proposed method. Table 2 represents the whole experiments in this paper, while Figure 3 and Table 1 only exhibit the same experiment results in different ways.

  $\bullet$	This paper lacks experiments on diverse datasets. The experiments in this paper only incorporate one dataset, which is unable to prove the effectiveness of the proposed method. Please consider at least three datasets in your experiments, such as Amazon [7] and Yelp [8].

  $\bullet$	This paper lacks sufficient baseline comparison. This paper only considers four baselines, which is insufficient.

  $\bullet$	This paper lacks ablation study. While this paper proposes several modules, the contribution of each module is unknown. Please conduct an ablation study to examine the effectiveness of each module.
4.	The paper writing of this paper is poor.

  $\bullet$		The implementation detail of the proposed method is unclear. While this paper proposes several modules, such as CF, CB, and CA, there is not any formula introducing how the model is constructed and trained. For example, in line 252, this paper states that the NER pre-trained model is fine-tune. However, the detail of fine-tuning is unknown. Please add more details about the implementation of the proposed method.

  $\bullet$		This paper overclaims its novelty. For example, in line 439, this paper claims that “this research advances the theoretical understanding”. However, there is not any theories in this paper.

  $\bullet$		The task formulation of tag recommendation is missing.

  $\bullet$		The introduction of baselines should not follow by the related work. Please refer to literature for where to introduce baselines [4, 8].
5.	The code is unavailable.


[1] Pennington, Jeffrey, Richard Socher, and Christopher D. Manning. "Glove: Global vectors for word representation." Proceedings of the 2014 conference on empirical methods in natural language processing (EMNLP). 2014.

[2] BehnamGhader, Parishad, et al. "Llm2vec: Large language models are secretly powerful text encoders." arXiv preprint arXiv:2404.05961 (2024).

[3] Neelakantan, Arvind, et al. "Text and code embeddings by contrastive pre-training." arXiv preprint arXiv:2201.10005 (2022).

[4] He, Xiangnan, et al. "Lightgcn: Simplifying and powering graph convolution network for recommendation." Proceedings of the 43rd International ACM SIGIR conference on research and development in Information Retrieval. 2020.

[5] Yu, Junliang, et al. "XSimGCL: Towards extremely simple graph contrastive learning for recommendation." IEEE Transactions on Knowledge and Data Engineering 36.2 (2023): 913-926.

[6] Wang, Shuhe, et al. "Gpt-ner: Named entity recognition via large language models." arXiv preprint arXiv:2304.10428 (2023).

[7] Hou, Yupeng, et al. "Bridging language and items for retrieval and recommendation." arXiv preprint arXiv:2403.03952 (2024).

[8] Asghar, Nabiha. "Yelp dataset challenge: Review rating prediction." arXiv preprint arXiv:1605.05362 (2016).

[9] Kang, Wang-Cheng, and Julian McAuley. "Self-attentive sequential recommendation." 2018 IEEE international conference on data mining (ICDM). IEEE, 2018.

**Questions:**

See in Weaknesses

---

### Official Review · Reviewer_sXFV · 2024-11-04

**Soundness:** 2
**Presentation:** 3
**Contribution:** 2
**Rating:** 5
**Confidence:** 4

**Summary:**

The authors present a novel approach for providing tag recommendations in a personalized, diverse, and context-aware manner. Their approach is a hybrid model that leverages Collaborative Filtering (CF), Content-Based (CB), and Context-Aware (CA) solutions.

They tested the solution on the MovieLens dataset, achieving approximately 0.81 in precision and 0.36 in recall. Several comparisons were presented. For their solution, they first apply CF, using GloVe vectors to derive tag embeddings. They then use k-means clustering on these embeddings, fine-tune a neural network model to determine the sentiment of the tags, and incorporate the sentiment scores into a user/movie matrix. Finally, they identify users similar to the current user to recommend tags. The solution also includes two other components: CB and CA.

In terms of suggestions, I would recommend that the authors expand the experimental section to include:
- Additional datasets (such as those referenced in Khan 2021, Xu 2023, Xin 2022, etc.).
- Additional comparisons, particularly since it is unclear why they did not compare their results to Khan 2021, which shows high recall. This is especially relevant given that the recall in this study is relatively low.

This is a highly competitive field with numerous existing works. Here are additional studies that the authors could consider for comparison:
- Studies in the “Survey on Collaborative Filtering, Content-Based Filtering, and Hybrid Recommendation Systems.”
- “Content-Based and Collaborative Techniques for Tag Recommendation: An Empirical Evaluation.”
- "Hybrid Recommendation System Based on Collaborative and Content-Based Filtering.”

To truly advance the state of the art, I encourage the authors to include more datasets and comparisons. Additionally, they may wish to explore solutions to improve recall. Starting with CF in the solution may be one reason for the low recall observed.

**Strengths:**

- Hybrid Recommendation Model: combining CF, CA, and CB. This integration of multiple methods demonstrates an effort to enhance the personalization, diversity, and context-awareness of tag recommendations, which could provide more accurate and relevant recommendations than relying on a single technique.
- Good performance in terms of precision

**Weaknesses:**

- Limited Dataset Variety: The authors only tested their solution on the MovieLens dataset, which may limit the generalizability of their results. Additional datasets:
- Amazon Product Review Dataset
- Last.fm Dataset
- Yelp Open Dataset
- Amazon instant video (if applicable from Khan 2021)
- musical instruments (if applicable from Khan 2021)
- office products (if applicable from Khan 2021)
- Low Recall and Lack of Comparisons with Key Studies: The recall achieved in this study is relatively low (~0.36), and the authors did not compare their results with other relevant works that report higher recall, specifically Khan 2021, Onan 2023, Zhang 2018.

**Questions:**

For me is important that you compare this solution with other works: Khan 2021, Onan 2023, Zhang 2018, I added additional reference in the previous sections.

Another important thing is to adopt additional open datasets:
- Yelp Open Dataset
- Amazon Product Review Dataset
- Last.fm Dataset
- Amazon instant video (if applicable from Khan 2021)
- musical instruments (if applicable from Khan 2021)
- office products (if applicable from Khan 2021)

---

### Note · Authors · 2024-11-21

**Comment:**

We have decided to withdraw our paper. Please ensure our paper remains confidential and is not disclosed.

**Withdrawal Confirmation:**

I have read and agree with the venue's withdrawal policy on behalf of myself and my co-authors.